# Soluble Urokinase Plasminogen Activator Receptor Levels Are Associated with Severity of Fibrosis in Patients with Primary Sclerosing Cholangitis

**DOI:** 10.3390/jcm11092479

**Published:** 2022-04-28

**Authors:** Burcin Özdirik, Martin Maibier, Maria Scherf, Jule Marie Nicklaus, Josephine Frohme, Tobias Puengel, Dirk Meyer zum Büschenfelde, Frank Tacke, Tobias Mueller, Michael Sigal

**Affiliations:** 1Department of Hepatology and Gastroenterology, Campus Virchow Klinikum (CVK) and Campus Charité Mitte (CCM), Charité Universitätsmedizin Berlin, Augustenburger Platz 1, 13353 Berlin, Germany; martin.maibier@charite.de (M.M.); maria.scherf@charite.de (M.S.); jule-marie.nicklaus@charite.de (J.M.N.); josephine.frohme@charite.de (J.F.); tobias.puengel@charite.de (T.P.); frank.tacke@charite.de (F.T.); tobias.mueller@charite.de (T.M.); michael.sigal@charite.de (M.S.); 2Berlin Institute of Health, 10178 Berlin, Germany; 3Charité—Universitätsmedizin Berlin, Corporate Member of Freie Universität Berlin, Humboldt-Universität zu Berlin, and Berlin Institute of Health, Institute of Laboratory Medicine, Clinical Chemistry and Pathobiochemistry, 13353 Berlin, Germany; dirk.meyer-zum-bueschenfelde@laborberlin.com; 4Labor Berlin—Charité Vivantes GmbH, 13353 Berlin, Germany; 5Berlin Institute for Medical Systems Biology (BIMSB), Max Delbrück Center for Molecular Medicine, 10115 Berlin, Germany

**Keywords:** primary sclerosing cholangitis, biomarkers, soluble urokinase plasminogen activator receptor, liver cirrhosis

## Abstract

The soluble urokinase-type plasminogen activator receptor (suPAR) has evolved as a useful biomarker for different entities of chronic liver disease. However, its role in patients with primary sclerosing cholangitis (PSC) is obscure. We analyzed plasma levels of suPAR in 84 patients with PSC and compared them to 68 patients with inflammatory bowel disease (IBD) without PSC and to 40 healthy controls. Results are correlated with clinical records. suPAR concentrations were elevated in patients with PSC compared to patients with IBD only and to healthy controls (*p* < 0.001). Elevated suPAR levels were associated with the presence of liver cirrhosis (*p* < 0.001) and signs of portal hypertension (*p* < 0.001). suPAR revealed a high accuracy for the discrimination of the presence of liver cirrhosis comparable to previously validated noninvasive fibrosis markers (area under the curve (AUC) 0.802 (95%CI: 0.702–0.902)). Further, we demonstrated that suPAR levels may indicate the presence of acute cholangitis episodes (*p* < 0.001). Finally, despite the high proportion of PSC patients with IBD, presence of IBD and its disease activity did not influence circulating suPAR levels. suPAR represents a previously unrecognized biomarker for diagnosis and liver cirrhosis detection in patients with PSC. However, it does not appear to be confounded by intestinal inflammation in the context of IBD.

## 1. Introduction

Primary sclerosing cholangitis (PSC) is a rare cholestatic liver disease associated with chronic inflammation of the biliary epithelium, resulting in multifocal bile duct strictures of the intra- and/or extrahepatic biliary tree and hepatic fibrosis. It ultimately leads to liver cirrhosis and end-stage liver disease [1,2,3]. In up to 80% of cases, a co-occurrence of inflammatory bowel disease (IBD) has been reported [1]. Despite significant progress in the molecular understanding of PSC, a major unmet need is the lack of established biomarkers for the accurate assessment of disease activity and progression [4,5]. Due to the low incidence of patients with PSC, analyses of potential biomarkers from well-characterized large patient cohorts are scarce. Alkaline phosphatase (ALP) and prognostic risk scores such as the Mayo risk score are frequently used as surrogate endpoints in clinical trials. However, they also have several limitations. For example, a spontaneous reduction in ALP levels can occur and does not necessarily reflect the disease state [6,7,8,9,10,11]. Moreover, a recent study reported no association between disease activity and ALP levels for over two years [11]. Therefore, at present, no established markers for reliable identification of changes in disease activity are available, which makes risk stratification and prognostication of high-risk patients an irrefutable problem.

Soluble urokinase-type plasminogen activator receptor (suPAR) is a soluble receptor derived from the cell surface receptor urokinase plasminogen activator receptor (uPAR), which is expressed by a variety of immune cells, including macrophages, neutrophils and activated T lymphocytes, as well as endothelial cells [12,13,14,15]. Interaction between the ligand urokinase plasminogen activator (uPA) and its receptor uPAR leads to cell migration, adhesion, proliferation and fibrinolysis [16,17]. uPA-deficient mice have a complex phenotype with defects in fibrinolysis, wound healing and neointima formation [18,19,20]. In the past years, altered circulating suPAR levels have been reported in different types of cancer as well as inflammatory and infectious diseases, such as sepsis, kidney and cardiovascular diseases [12,21,22,23,24,25,26]. Moreover, elevated suPAR levels were described in many acute and chronic liver diseases, suggesting its application as a valuable biomarker for risk stratification in chronic liver disease of different etiologies [20,27,28,29,30,31]. However, in patients with PSC the role of circulating suPAR as a potential biomarker has not yet been comprehensively elucidated.

Therefore, in this study, we investigated the potential role of circulating suPAR as a biomarker in a well-characterized and large cohort of patients with PSC. Since the majority of patients with PSC suffer concomitantly from IBD, we additionally performed an analysis in a cohort of patients that suffer from IBD but do not have PSC.

## 2. Patients and Methods

### 2.1. Study Design and Patient Cohort

In this cross-sectional prospective study, we analyzed circulating levels of suPAR as a novel diagnostic marker in a cohort of 84 patients with PSC and 68 patients with IBD only. The IBD cohort included 29 patients with ulcerative colitis (UC) and 39 patients with Crohn’s disease (CD). We compared our results to 40 healthy controls. From June 2021 until recently, we prospectively recruited patients who were admitted to our hepatology and gastroenterology unit and specialized outpatient clinics for autoimmune liver diseases and IBD at Charité Universitätsmedizin Berlin. Patients with small-duct PSC and PSC patients with previous transplantations were excluded. Diagnosis of PSC was confirmed in accordance with the current American College of Gastroenterology guidelines, using a combination of clinical, biochemical and cholangiographic (magnetic resonance cholangiopancreatography and/or endoscopic retrograde cholangiopancreatography) features [32,33,34,35]. Acute cholangitis was defined according to the revised Tokyo guidelines 2018 based on a combination of systemic inflammation (fever and/or shaking chills, increase in serum C-reactive protein levels, abnormal white blood cell counts), cholestasis (jaundice/abnormal liver function tests) and imaging (biliary dilatation, evidence of the etiology on imaging (stricture, stone, stent, etc.) [36]. Only cholangitis episodes necessitating hospitalization and intravenous antibiotic treatment at time of suPAR analysis were included in our analysis. For diagnosis of autoimmune hepatitis (AIH), simplified diagnostic criteria for AIH in clinical practice were used. Patients were categorized as overlap PSC/AIH when fulfilling the diagnostic criteria for AIH and PSC. Diagnosis of hepatobiliary malignancies was confirmed histologically. Dominant stenosis was defined as stenosis measuring <1.5 mm in the common bile duct or <1.0 mm in the hepatic ducts at cholangiography [37,38]. Detection of fibrosis stage was carried out using liver stiffness measurement by FibroScanMini430 (Echosens; Paris, France) according to current guidelines [39]. The automatic probe selection tool included in the device’s software automatically measured the probe-to-liver capsule distance and indicated the probe (M or XL) to be used according to the patient’s morphology. Cut-off values for fibrosis stages F0–F1, F2, F3 and F4 were <8, 7 kPa, ≥8, 8 kPa, ≥9, 6 kPa and ≥14, 4 kPa, respectively. Liver cirrhosis was diagnosed based on liver stiffness measurement (cut-off ≥ 14.4 kPa) or presence of (morphological) signs of hepatic decompensation such as portal hypertension (ascites, esophageal varices) and/or hepatic encephalopathy [39,40]. For comparison of the diagnostic accuracy in estimating the severity of fibrosis, blood-based biomarkers, such as fibrosis-4 (FIB4) index and aspartate transaminase-to-platelets ratio index (APRI) were used [41,42]. The Mayo risk score, model of end-stage liver disease (MELD) and Child–Pugh score (CHILD) were calculated using published algorithms [43,44,45,46,47]. Definition of IBD was based on endoscopic and histological findings according to accepted criteria [48,49,50]. Patients with colitis indeterminata (*n* = 2) were excluded from our study. Disease activity in patients with IBD was based on Mayo score/disease activity index for UC and Harvey–Bradshaw index for CD [51,52]. Extraintestinal manifestations in patients with IBD included arthralgia, uveitis, erythema nodosum, aphtous ulcers, pyoderma gangrenosum, anal fissures, fistulas and abscesses (based on the Harvey–Bradshaw index for CD).

### 2.2. Measurement of Circulating suPAR Levels

Patients’ blood samples were collected and centrifuged for 10 min at 2000× *g*, and heparin plasma aliquots of 1 mL were frozen immediately at −80 °C to avoid repetitive freeze–thaw cycles until use. Plasma levels of suPAR were measured with suPARnostic^®^ TurbiLatex test (Nr. T004, suPARnostic, ViroGates, Birkerød, Denmark) on a Cobas c501/502 clinical chemistry analyser (Roche Diagnostics Ltd., Burgess Hill, UK) at Labor Berlin, the central laboratory of Charité, University Medicine Berlin, Germany. suPARnostic^®^ TurbiLatex test is a turbidimetric immunoassay that quantitatively measures suPAR in human plasma samples. In detail, 10 µL of heparin plasma was mixed with 150 μL of dilution buffer. After an incubation period of 5 min, 50 μL of a suspension of latex particles coated with rat and mouse monoclonal antibodies to suPAR was added, and suPAR aggregation started. The level of accumulation was defined by the amount of scattered light during measurement of light absorption. Linear regression was used to evaluate the absorbance values and calculate the plasma concentration.

Standard laboratory parameters, such as hemoglobin, leucocytes, platelets, total bilirubin, creatinine, aspartate aminotransferase (AST), alanine aminotransferase (ALT), ALP, gamma-glutamyl transferase (GGT), C-reactive protein (CRP), albumin and international normalized ratio (INR), were measured at Labor Berlin, the central laboratory of Charité, University Medicine Berlin, Germany. suPAR concentrations were linked to the patients’ clinical characteristics and correlated to their outcome. Clinical and demographic data including laboratory parameters were extracted from electronic medical charts.

### 2.3. Statistical Analysis

Nonparametric data were compared using the Mann–Whitney *U*-test or the Kruskal–Wallis test for multiple group comparisons. Frequencies were compared using the χ^2^ test or the Fisher’s exact test, where appropriate. Box plots are displayed, where the bold line indicates the median per group, and the box represents 50% of the values. The edges of the box are the first and third quartiles. The horizontal lines show minimum and maximum values of calculated nonoutlier values. The dots represent calculated outliners. Nonparametric Spearmen’s rho (two-tailed) correlation was performed to analyze the correlation between suPAR and continuous variables. We generated receiver operating characteristic (ROC) curves by plotting the sensitivity (%) against 100% specificity (%). Youden index method (YI = sensitivity + specificity − 1) was used for calculation of optimal cut-off values for ROC curves. All statistical analyses were performed with SPSS (Version 26.0. Armonk, NY, USA: IBM Corp). A *p*-value of <0.05 was considered statistically significant (* *p* < 0.05; ** *p* < 0.01; *** *p* < 0.001).

### 2.4. Consent

The study protocol was reviewed and approved by the institutional ethics committee (ethical approval number EA1/142/21) and was performed in accordance with the Declaration of Helsinki.

## 3. Results

### 3.1. Baseline Patient and Clinical Characteristics—PSC Cohort

Our final cohort comprised 84 patients with confirmed PSC, which were compared to 68 patients with confirmed IBD and 40 healthy controls.

The median age of patients with PSC at the time of suPAR analysis was 45 years (20–71). The median age at initial diagnosis of PSC was 31 years (9–67). Thirty-three percent of the patients (*n* = 28) were female, and sixty-seven percent (*n* = 56) were male. Fourteen patients (17%) had been diagnosed with an overlap syndrome with autoimmune hepatitis. Sixty patients (71%) suffered concomitantly from IBD; of these, forty-nine patients (58%) had UC and six patients (7%) had a diagnosis of CD, while four patients (5%) showed evidence of both diagnoses. Seventy-eight patients (93%) were treated with ursodeoxycholic acid (UDCA). Baseline patient and clinical characteristics of the PSC cohort are shown in Table 1.

### 3.2. Circulating suPAR Is Elevated in Patients with PSC

First, we compared levels of circulating suPAR in plasma of 84 patients with PSC and 40 healthy controls. Our data revealed increased suPAR plasma concentrations in the PSC cohort compared to healthy controls (*p* < 0.001) (Figure 1A). For quantification of the discriminatory power of suPAR to distinguish between PSC and healthy, we performed receiver operating characteristic (ROC) curve analyses and revealed an area under the curve (AUC) of 0.724 (95% confidence interval (CI): 0.641–0.816). At the optimal cut-off value of 3.96 ng/mL, suPAR showed a sensitivity of 67% and 73% for identification of PSC (Figure 1B).

### 3.3. suPAR Levels in PSC Patients Are not Associated with Metabolic Risk Factors or Comorbidities

Since suPAR levels in patients with PSC compared to healthy controls revealed such a striking difference, we hypothesized that suPAR levels might be indicative of specific patient characteristics. Since alterations in suPAR concentrations have been already reported in the context of cardiovascular and metabolic diseases, we analyzed whether the presence of age >60, body mass index (BMI) >24 kg/m^2^, arterial hypertension, diabetes mellitus and smoking might influence plasma suPAR levels in patients with PSC [53,54,55]. Of note, in our study cohort, suPAR concentrations were not associated with the presence of these risk factors or medical conditions (Appendix A), which might be related to the young age of our PSC patient cohort and PSC patients in general. Furthermore, levels of circulating suPAR were not associated with gender (Appendix A).

### 3.4. Increased suPAR Levels in PSC Patients Are Associated with the Presence of Liver Cirrhosis

Further analysis revealed a strong association of suPAR levels with the presence of liver cirrhosis. SuPAR levels were increased in patients with PSC and liver cirrhosis compared to patients with PSC and without liver cirrhosis (*p* < 0.001) (Figure 2A). Moreover, an increase with higher CHILD stages (*p* = 0.11) and MELD scores could be detected (*p* < 0.001; r = 0.547) (Figure 2B,C). Figure 2D displays the increase in circulating suPAR levels with increasing fibrosis stage (*p* = 0.017; F0 vs. F4 *p* = 0.001). ROC curve analysis for suPAR revealed an AUC value of 0.802 (95%CI: 0.702–0.902) for the discrimination between PSC patients with and without liver cirrhosis. We applied the Youden index method and generated an ideal cut-off value of 5.35 ng/mL, at which suPAR plasma levels demonstrated a diagnostic sensitivity and specificity of 85% and 73%, respectively. To analyze suPAR’s diagnostic accuracy in detecting liver cirrhosis, we compared it to non-disease-specific blood-based biomarkers such as the FIB4 index and APRI score [39]. SuPAR revealed an AUC comparable to the FIB4 index (AUC 0.836; 95%CI (0.752–0.920) and APRI score (AUC 0.866; 95%CI (0.789–0.944). Notably, combining circulating suPAR and the APRI score even further improved the diagnostic power for the discrimination of the presence of liver cirrhosis with a sensitivity of 89% and specificity of 77% (AUC 0.859 (95%CI (0.780–0.938)) (Figure 2E). The subdivision of our cohort into patients with no to mild fibrosis (F0–F2) and patients with a higher degree of fibrosis (F3–F4) demonstrated a lower AUC of 0.771 (95%CI (0.669–0.874) for suPAR. However, the AUC for suPAR was still comparable to the composite scores, i.e., the FIB4 index (AUC 0.812; 95%CI (0.720–0.905) and APRI score (AUC 0.844; 95%CI (0.754–0.935).

Consistent with our results, further analysis with Spearman rank correlation revealed an association between suPAR and the FIB4 index (*p* < 0.001; r = 0.576) and liver stiffness (*p* < 0.001; r = 0.581) (Appendix A). In line with this, liver synthesis parameters, such as bilirubin (*p* < 0.001; r = 0.525), albumin (*p* < 0.001; r = −0.638), platelets (*p* < 0.001; r = −0.413) and prothrombin (*p* < 0.001; r = −0.5), correlated with suPAR levels (Appendix A). Moreover, patients with liver cirrhosis and apparent portal hypertension, such as ascites, esophageal varices or splenomegaly, revealed higher suPAR levels compared to patients without signs of portal hypertension (*p* < 0.001). Analysis of these particular factors individually revealed higher suPAR levels in patients with ascites (*p* < 0.001), esophageal varices (*p* < 0.001) and splenomegaly (*p* = 0.03) (Appendix A).

### 3.5. suPAR Is Elevated in PSC Patients with Acute Cholangitis but Does Not Indicate the Presence of Dominant Stenosis

The clinical course of progressive PSC is characterized by recurrent episodes of cholangitis, which is a complication of PSC that is closely linked to an increased risk for hepatic decompensation [56]. Our analysis revealed a significant elevation in suPAR levels in patients with acute cholangitis episodes (*p* < 0.001) (Figure 3A). In line with this, Spearman rank correlation showed a significant positive correlation between circulating suPAR levels and CRP values (*p* < 0.001; r = 0.641) (Figure 3B). Since up to 50% of patients with PSC develop inflammatory strictures in the biliary tree, which cause jaundice and bacterial cholangitis as well as predilection spots for development of neoplastic transformation, we next hypothesized that the presence of dominant stenoses (detected via endoscopic retrograde cholangiography–pancreatography (ERCP) and/or magnetic resonance cholangiopancreatography (MRCP) imaging) might be linked to higher suPAR levels [57,58,59,60]. However, suPAR levels of patients with and without dominant stenosis displayed similar suPAR concentrations (Figure 3C).

### 3.6. suPAR Correlates with Prognostic Parameters in PSC Patients

Further, we performed additional Spearman rank correlation and compared suPAR to the most widely used prognostic biomarkers, ALP and the Mayo risk score, revealing a significant positive correlation for both parameters (Mayo risk score: *p* < 0.001, r = 0.649; ALP: *p* = 0.001; r = 0.343) (Appendix A).

### 3.7. suPAR Levels Are not Elevated in Patients with IBD and Do Not Reflect Disease Activity

In the next step, we analyzed whether the presence of overlap syndrome or IBD is linked to suPAR levels in PSC patients, revealing similar suPAR levels in both cohorts (Appendix A). Considering the high number of patients with PSC suffering concomitantly from IBD (*n* = 60; 71%), we aimed to find out whether plasma suPAR concentrations are elevated in patients with and without IBD. Therefore, we compared suPAR levels in 68 patients with IBD only (CD: *n* = 39; UC: *n* = 29) to suPAR levels in 40 healthy controls. Our IBD cohort comprised 44 (65%) male and 24 (35%) female patients with a median age of 42 years (18–88 years). The main IBD manifestation sites were the colon (*n* = 39; 57%), ileum only (*n* = 19; 28%), ileocolon (*n* = 6; 9%), the small intestine (*n* = 3; 4%) and the upper gastrointestinal system (*n* = 1; 2%). Twenty-eight percent of our patients with IBD developed extraintestinal manifestations. Baseline patient and clinical characteristics as well as laboratory parameters of the IBD cohort are outlined in Table 2 and Appendix A.

Our data demonstrated similar suPAR levels in patients with IBD compared to healthy controls regardless of the presence of CU or CD, respectively (Figure 4A,B). Moreover, suPAR concentrations were not linked to the IBD manifestation side nor to the disease activity in patients with IBD and CU and CD, respectively (Figure 4C–E). In the next step, we performed Spearman rank correlation of suPAR levels in patients with CD and UC, revealing a positive correlation between suPAR and CRP levels in patients with CD (*p* = 0.003; r = 0.473) (Figure 5A). Our analysis showed no link between suPAR and CRP levels in patients with UC (r = 0.315; *p* = 0.096) (Figure 5B). SuPAR levels were not linked to calprotectin levels in patients with CD and UC, respectively (Figure 5C,D).

Taken together, here we present, in a cross-sectional study comprising a well-characterized and large cohort of patients with PSC, that plasma levels of suPAR are elevated in patients with PSC and reflect the degree of fibrosis. Further, we demonstrate that suPAR levels can indicate the presence of acute cholangitis episodes. However, despite the high proportion of PSC patients with IBD, the presence of IBD and its IBD disease activity did not influence suPAR levels.

## 4. Discussion

Despite the progressive nature of PSC, there have only been a few advances in the establishment of diagnostic and prognostic biomarkers. To our knowledge, the present study is the first to evaluate suPAR as a biomarker in a well-characterized large cohort of patients with PSC, using patients with IBD as a control cohort. Previously, Loosen et al. compared suPAR levels in patients with biliary tract cancer to eleven patients with PSC who showed no evidence of cancer. In line with our results, suPAR levels in patients with PSC were increased compared to healthy controls, but lower than in patients with biliary tract cancer [61].

We report that suPAR could represent a promising candidate for fibrosis detection in patients with PSC since we revealed that a higher degree of fibrosis is linked to higher suPAR levels. Moreover, the specificity and sensitivity of suPAR were comparable to well-established fibrosis scores such as the FIB4 index and APRI score. Consistent with our findings, previous studies have already pointed out that systemic suPAR levels correlate with liver fibrosis in the absence of systemic inflammation with different entities of chronic liver diseases, including nonalcoholic fatty liver disease (NAFLD), hepatitis B, hepatitis C and alcoholic liver disease [20,27,28,30,62,63]. The identification of individuals with severe liver fibrosis among patients with chronic liver disease is crucial for the accurate evaluation of disease activity, disease progression, and risk-stratification as well as therapy decisions. At present, percutaneous liver biopsy is still considered the most important diagnostic tool for the staging of liver fibrosis. However, it is not only limited by significant rates of complications, but also by the subjectivity of interpretation as well as a sampling variability of 20% [4,10,64]. Noninvasive methods such as measurement of liver stiffness are helpful, but their use is time-consuming and their availability often limited to hepatologic centers. Thus, further reliable noninvasive markers of liver fibrosis are needed [39,62].

From a molecular perspective, liver fibrosis is the result of a continuous inflammatory process and an imbalance between the production of the extracellular matrix and its degradation. The urokinase plasminogen activator receptor (uPAR) is expressed on most leucocytes, which play a pivotal role in the pathogenesis of hepatic inflammation and fibrosis, including neutrophils, lymphocytes, monocytes and macrophages [16,17]. In particular, activated monocytes and liver-resident macrophages are assumed to be the major source of circulating suPAR in patients with cirrhosis even in the absence of overt infection [29,65]. uPA-deficient mice have a complex phenotype with defects in fibrinolysis, wound healing and neointima formation [18,19,20]. May et al. found that beta 2 integrin-mediated adhesion of leukocyte–endothelial cell interactions and recruitment to inflamed areas require the presence of uPAR [66]. Notably, mice deficient in uPA and uPAR were protected from hepatic fibrosis in experimental liver injury models, probably due to immunomodulatory actions of uPA in hepatic fibrogenesis. Based on their experimental studies involving uPA and uPAR knock-out mice, Higazi et al. concluded that plasminogen activators affect fibrosis partly by liver-specific activation of CD8 cells, which regulate the fibrogenic activity of hepatic stellate cells [20,67]. However, further investigations, including human models, are needed to determine whether suPAR represents an epiphenomenon or if it plays an active role in the development of liver fibrosis.

suPAR is considered as a biomarker with little circadian changes in plasma concentrations and high stability in serum samples [68,69,70,71]. In our study, suPAR levels were independent of patients’ age and sex, highlighting the stability of this marker in the context of PSC. Contrary to previous studies, which revealed a link between elevated suPAR levels and cardiovascular diseases, diabetes, age and even sex in the general population, we could not find a link between suPAR levels and these particular factors [54]. In line with our results, Sjöwall and colleagues showed no association between the presence of diabetes, gender or a higher BMI in patients with chronic hepatitis C [62]. However, they could demonstrate that suPAR levels were age-dependent in patients with hepatitis C and NAFLD [62].

A prognostically relevant risk factor of PSC is the occurrence of recurrent and potentially lethal cholangitis episodes, which are triggered by bile stasis caused by the presence of a dominant stenosis in the bile ducts. We found that suPAR levels correlate with CRP levels in patients with PSC and that suPAR levels can indicate the presence of acute cholangitis episodes, which is consistent with previous findings suggesting the involvement of suPAR in systemic inflammatory responses [71]. Moreover, our data revealed that suPAR correlates with prognostic markers such as ALP and the Mayo risk score. However, the prognostic value of suPAR in PSC patients has not been analyzed yet and makes further investigation necessary. Prospective clinical trials, comparing the prognostic value of suPAR in respect to surrogate endpoints such as transplant-free survival, death, hepatic decompensation and development of malignancies should be performed in the near future.

To our knowledge, at present, there are only two small-sized studies (*n* < 40) that have investigated suPAR levels in IBD patients. Our data demonstrated similar suPAR levels in patients with IBD compared to healthy controls, neither reflecting the disease activity nor the IBD manifestation side. These results are in line with a previous study in pediatric IBD patients [72]. In contrast, Lönnkvist et al. found elevated suPAR levels in adult patients with CD. Interestingly, these levels remained unchanged after treatment initiation with infliximab [73]. CRP is one of the few biomarkers that has shown predictive qualities regarding severe CD ileitis [74]. In our study, CRP levels in patients with CD but not CU correlated with suPAR levels. This is not surprising since CRP has shown predictive qualities in severe CD ileitis [75]. Moreover, suPAR levels in plasma were comparable in CD and UC, mirroring the fact that most patients with CD showed colonic or ileocolonic disease [72]. Despite the ability of suPAR to reflect disease activity in the liver, our results corroborate the notion that suPAR is not an accurate marker of intestinal inflammatory activity in the context of IBD. However, due to contradictory results in previous studies, large-sample-sized longitudinal studies are needed to better address this question.

The main limitations of our prospective cross-sectional study were the monocenter trial design and the lack of longitudinal measurements for further analysis of suPAR fluctuations, e.g., during endoscopic treatment. In addition, in our study, fibrosis detection was based only on liver stiffness measurements and not on a liver biopsy, which is still considered the gold standard for fibrosis detection.

In conclusion, our data suggest that suPAR might be a clinically useful diagnostic marker for patients with PSC and, in particular, for the detection of fibrosis stage and acute cholangitis episodes in patients with PSC. Moreover, we report that suPAR is not confounded by the presence or disease activity of IBD. Our results warrant both prospective cross-sectional and longitudinal clinical studies with larger patient cohorts as well as experimental studies to further explore suPAR as a biomarker and its specific effects and regulatory mechanisms in patients with PSC.

## Figures and Tables

**Figure 1 jcm-11-02479-f001:**
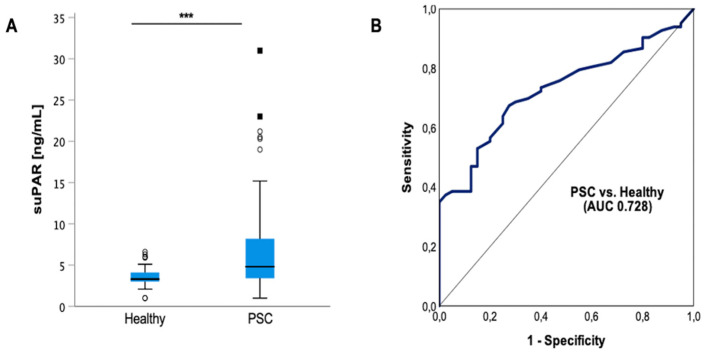
**Plasma suPAR levels are elevated in patients with PSC**. Circulating soluble urokinase-type plasminogen activator receptor (suPAR) levels are elevated in patients with PSC compared to healthy controls (*p* < 0.001). Box plots are displayed, where the bold line indicates the median per group, and the edges of the box are the first and third quartiles. The horizontal lines show minimum and maximum values of calculated nonoutlier values. The dots represent calculated outliners (**A**). Application of receiver operating characteristic (ROC) curve analyses to quantify the discriminatory power of suPAR revealed an area under the curve (AUC) of 0.728 (95% confidence interval (CI): 0.641–0.816) (**B**). Soluble urokinase-type plasminogen activator receptor (suPAR). Area under the curve (AUC). (*** *p* < 0.001).

**Figure 2 jcm-11-02479-f002:**
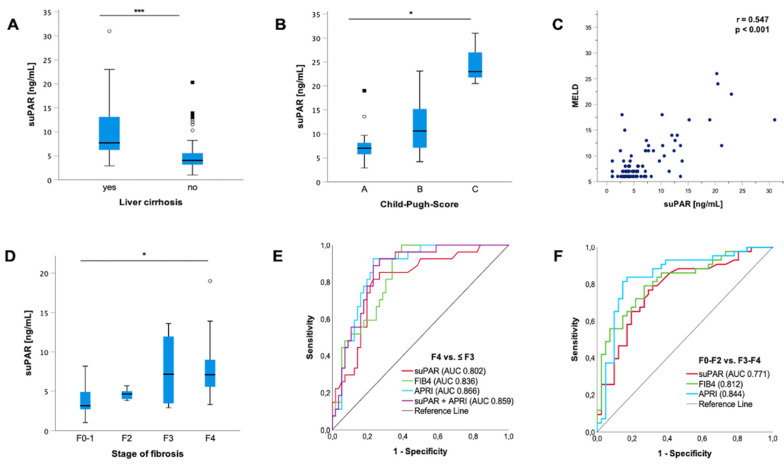
**suPAR is associated with the presence of liver cirrhosis.** Soluble urokinase-type plasminogen activator receptor (suPAR) is elevated in patients with liver cirrhosis (*p* < 0.001) (**A**). suPAR is linked to an increasing Child–Pugh score (CHILD) in patients with liver cirrhosis (*p* = 0.019; A–C: *p* = 0.047; B–C: *p* = 0.046; A–B: *n*.s.) (**B**). In line with this, according to Spearman rank correlation, suPAR increases with increasing model of end-stage liver disease (MELD) score (*p* > 0.001; r = 0.547) (**C**). Subdivision of groups with different fibrosis stages revealed an elevation in suPAR levels (*p* = 0.017; F1 vs. F4 *p* = 0.001; F1–F2: *n.*s.; F1–F3: *n.*s.;F2–F3: *n.*s.) (**D**). suPAR revealed a high accuracy for the discrimination of the presence of liver cirrhosis (area under the curve (AUC) 0.802 (95%CI: 0.702–0.902), which is comparable to previously validated noninvasive fibrosis markers such as the fibrosis-4 (FIB4) index (AUC 0.836; 95%CI (0.752–0.920) and aspartate transaminase-to-platelets ratio index (APRI) score (AUC 0.866; 95%CI (0.789–0.944). The combination of suPAR and APRI score further improved the diagnostic power for the discrimination of the presence of liver cirrhosis (AUC 0.859 (95%CI (0.780–0.938)) (**E**). Subdivision of our cohort into patients with no to mild fibrosis (F0–F2) and patients with a higher degree of fibrosis (F3–F4) demonstrated a lower AUC of 0.771 (95%CI (0.669–0.874) for suPAR. However, the AUC was still comparable to the FIB4 index (AUC 0.812; 95%CI (0.720–0.905) and APRI score (AUC 0.844; 95%CI (0.754–0.935) (**F**). Box plots are displayed, where the bold line indicates the median per group and the edges of the box are the first and third quartiles. The horizontal lines show minimum and maximum values of calculated nonoutlier values. The dots represent calculated outliners. Soluble urokinase-type plasminogen activator receptor (suPAR). Child–Pugh Score (CHILD). Model of end-stage liver disease (MELD). Fibrosis-4 (FIB4) index. Aspartate transaminase-to-platelets ratio index (APRI). (* *p* < 0.05; *** *p* < 0.001).

**Figure 3 jcm-11-02479-f003:**
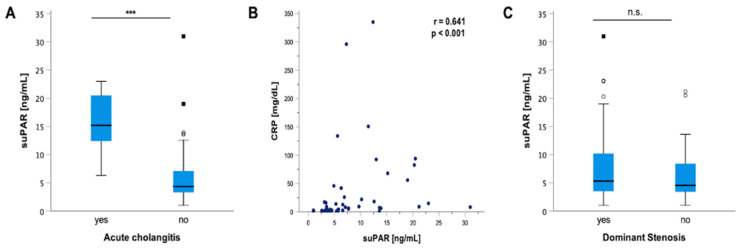
**suPAR levels indicate acute cholangitis but not the presence of a dominant stenosis.** Our analysis revealed a significant elevation in soluble urokinase-type plasminogen activator receptor (suPAR) levels in patients with acute cholangitis episodes compared to patients without acute cholangitis (*p* < 0.001) (**A**). In line with this, Spearman rank correlation showed a positive correlation between circulating suPAR levels and CRP values (*p* < 0.001; r = 0.641) (**B**). However, circulating suPAR levels did not indicate the presence of dominant stenosis (**C**). Box plots are displayed, where the bold line indicates the median per group and the edges of the box are the first and third quartiles. The horizontal lines show minimum and maximum values of calculated nonoutlier values. The dots represent calculated outliners. Soluble urokinase-type plasminogen activator receptor (suPAR). (*** *p* < 0.001).

**Figure 4 jcm-11-02479-f004:**
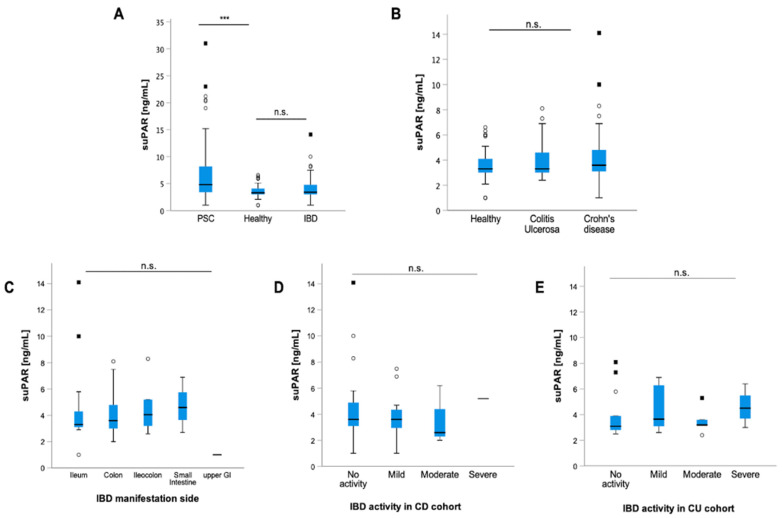
**suPAR levels are not increased in patients with inflammatory bowel disease.** Compared to healthy controls, soluble urokinase-type plasminogen activator receptor (suPAR) levels were elevated in patients with PSC (*p* < 0.001), but not in patients with inflammatory bowel disease (IBD) (**A**). Moreover, suPAR levels were independent of the presence of colitis ulcerosa (CU) and Crohn’s disease (CD) (**B**). suPAR levels did not differ depending on the IBD manifestation side (**C**), and they did not reflect the disease activity in patients with CU (**D**) and CD, respectively (**E**). Box plots are displayed, where the bold line indicates the median per group and the edges of the box are the first and third quartiles. The horizontal lines show minimum and maximum values of calculated nonoutlier values. The dots represent calculated outliners. Soluble urokinase-type plasminogen activator receptor (suPAR). Colitis ulcerosa (CU) and Crohn’s disease (CD). (*** *p* < 0.001).

**Figure 5 jcm-11-02479-f005:**
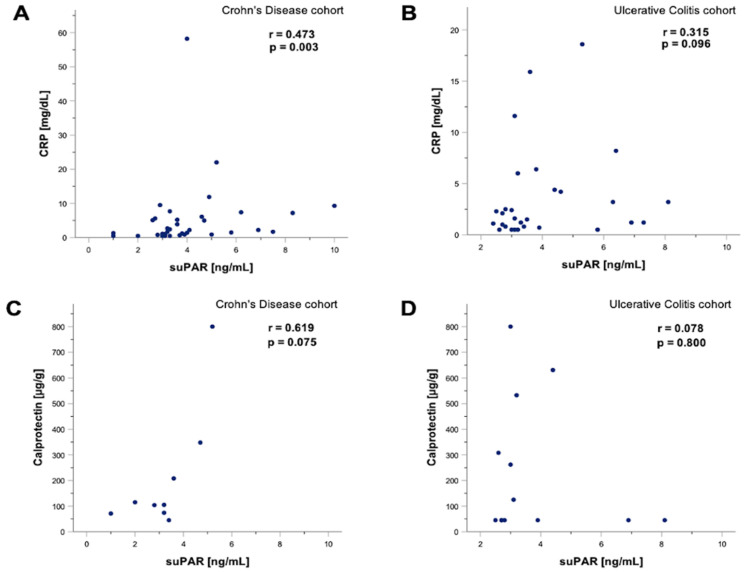
**suPAR levels correlate with CRP levels in patients with CD.** Soluble urokinase-type plasminogen activator receptor (suPAR) levels correlated with C-reactive protein (CRP) in patients with Crohn’s disease (CD). Spearman rank correlation revealed a positive correlation between suPAR and CRP levels in patients with CD (*p* = 0.003; r = 0.473), but not in patients with ulcerative colitis (UC) (*p* = 0.096; r = 0.315) (**A**,**B**). Spearman rank correlation of suPAR and fecal calprotectin levels in patients with UC demonstrated no significant association; however, a tendency towards a significant positive correlation could be detected (*p* = 0.075; r = 0.619). In contrast, suPAR levels appeared to be independent of fecal calprotectin levels in patients with CD (*p* = 0.075; r = 0.619) and CU (*p* = 0.8; r = 0.078) (**C**,**D**). Soluble urokinase-type plasminogen activator receptor (suPAR). Gastrointestinal (GI). Colitis ulcerosa (CU). Crohn’s disease (CD). C-reactive protein (CRP).

**Table 1 jcm-11-02479-t001:** Baseline patient and disease characteristics of the PSC cohort.

	PSC Cohort *n* = 84
**Gender**	
▪ Female	28 (33%)
▪ Male	56 (67%)
**Median current age (range)**	45 (20–71)
▪ Age < 60	71 (85%)
▪ Age ≥ 60	13 (15%)
**Median age at initial diagnosis (range)**	31 (9–67)
**Mean BMI** [kg/m^2^] (SD)	24 ± 3.3
▪ BMI < 24	36/69 (52%)
▪ BMI ≥ 24	33/69 (48%)
**Alcohol**	
▪ Yes	7 (8%)
▪ No	77 (92%)
**Smoking**	
▪ Yes	7 (8%)
▪ No	77 (92%)
**Comorbidities**	
▪ Arterial hypertension	16 (19%)
▪ Diabetes	2 (2%)
▪ Asthma/COPD	3 (4%)
**Overlap syndrome with AIH**	
▪ Yes	14 (17%)
▪ No	70 (83%)
**Presence of IBD**	60 (71%)
▪ Ulcerative colitis	50 (60%)
▪ Crohn’s disease	6 (7%)
▪ Undetermined	4 (5%)
**Presence of liver cirrhosis**	
▪ Yes	27 (32%)
▪ No	57 (68%)
**CHILD classification**	
▪ A	16/27 (59%)
▪ B	8/27 (30%)
▪ C	3/27 (11%)
**Portal hypertension**	
▪ Yes	24 (29%)
▪ No	60 (71%)
**Esophageal varices**	
▪ No	67 (80%)
▪ Grade I	10/17 (59%)
▪ Grade II	3/17 (18%)
▪ Grade III	4/17 (24%)
**Ascites**	
▪ Yes	11 (13%)
▪ No	73 (87%)
**Acute Cholangitis**	
▪ Yes	9 (11%)
▪ No	75 (89%)
**Dominant Stenosis**	
▪ Yes	53 (63%)
▪ No	31 (37%)
**UDCA treatment**	
▪ Yes	78 (93%)
▪ No	6 (7%)

Data are *n* (%) of patients, if not indicated otherwise. The percentages were rounded and may not total 100%. Significant results (*p* < 0.05) are shown in bold type. Primary sclerosing cholangitis (PSC), body mass index (BMI); standard deviation (SD); chronic obstructive pulmonary disease (COPD); autoimmune hepatitis (AIH); inflammatory bowel disease (IBD); Child–Pugh Score (CHILD), ursodeoxycholic acid (UDCA).

**Table 2 jcm-11-02479-t002:** Baseline patient and disease characteristics of the IBD cohort.

	All IBD Patients *n* = 68	Crohn‘s Disease *n* = 39	Ulcerative Colitis *n* = 29	*p*-Value
**Gender**				0.555
▪ Female	24 (35%)	14 (36%)	10 (35%)	
▪ Male	44 (65%)	25 (64%)	19 (66%)	
**Median current age (range)**	42 (18–88)	48 (18–86)	38 (22–88)	0.624
**Median age at initial diagnosis (range)**	30 (13–77)	30 (13–77)	27 (13–62)	0.827
**Mean BMI** (SD)	24 ± 4,9	24 ± 5.3	24 (16–31)	0.442
**Comorbidities**				
▪ Arterial hypertension	15 (21%)	10 (26%)	5 (17%)	0.301
▪ Diabetes	3 (4%)	1 (3%)	2 (7%)	0.389
▪ Asthma/COPD	9 (13%)	5 (13%)	4 (14%)	0.591
**Extraintestinal Manifestation**				0.373
▪ Yes	19 (28%)	12 (31%)	7 (24%)	
▪ No	51 (73%)	27 (69%)	22 (76%)	
**Disease activity at time of analysis**				0.169
▪ No disease activity	36 (53%)	20 (51%)	16 (55%)	
▪ Yes	32 (47%)	19 (49%)	13 (45%)	
▪ Mild	18 (27%)	14 (36%)	4 (14%)	
▪ Moderate	12 (18%)	4 (10%)	8 (28%)	
▪ High	2 (3%)	1 (3%)	1 (3%)	
**IBD manifestation side**				**<0.001**
▪ Upper GI tract (esophagus, duodenum)	1 (2%)	1 (3%)		
▪ Small intestine (duodenum, jejunum, ileum)	3 (4%)	3 (8%)		
▪ Ileum only	19 (28%)	19 (49%)		
▪ Ileocolon	6 (9%)	5 (13%)	1 (3%)	
▪ Colon	39 (57%)	11 (28%)	28 (97%)	
**Surgery**				**0.008**
▪ Yes	21 (31%)	17 (44%)	4 (14%)	
▪ No	47 (69%)	22 (56%)	25 (86%)	
**Development of colorectal carcinoma**				0.325
▪ Yes	2 (3%)	2 (5%)	0 (0%)	
▪ No	56 (97%)	27 (95%)	29 (100%)	

Data are *n* (%) of patients, if not indicated otherwise. The percentages were rounded and may not total 100%. Significant results (*p* < 0.05) are shown in bold type. Inflammatory bowel disease (IBD). Gastrointestinal (GI).

## Data Availability

Data are available upon request from the Department of Hepatology and Gastroenterology of the Charité University Medicine Berlin for researchers who meet the criteria for access to confidential data (gastro-cvk@charite.de).

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
