# Peer review of "Soluble Urokinase Plasminogen Activator Receptor Levels Are Associated with Severity of Fibrosis in Patients with Primary Sclerosing Cholangitis"

_jcm, 2022, doi:10.3390/jcm11092479_

Round 1
Reviewer 1 Report
This is a well-organized study about the role of the diagnostic marker for the detection of the grade of liver fibrosis for PSC patients. However, there are also some issues that the authors need to address.
- Results
Page 7, line 197: Since supar levels in patients with PSC compared to healthy controls revealed such a striking difference,
- Since suPAR levels in patients with PSC compared to healthy controls revealed such a striking difference,
The figures inserted in the body are fuzzy overall and are not well visible in the PDF file version. If possible, please submit in vector file format.
Page 9, line 273: 3.5. suPAR is elevated in PSC patients with acute cholangitis, but does not indicate the presence of dominant stenosis
- Please clarify the definition of dominant stenosis in the method section. For example, Bismuth classification for benign biliary stricture.
Author Response
We are grateful for the positive feedback of the reviewer.
1. We corrected the following sentence as suggested by the reviewer:
"Since suPAR levels in patients with PSC compared to healthy controls revealed such a striking difference," (p.7; l. 229)
2. Reviewer: "The figures inserted in the body are fuzzy overall and are not well visible in the PDF file version. If possible, please submit in vector file format."
Response: We thank the reviewer for this important comment. We prepared the figures in a PDF format and uploaded them separately. Since the revised version has to be uploaded as a word document, we did not include the figures in PDF format into the word version yet.
3. Reviewer: Page 9, line 273: ." suPAR is elevated in PSC patients with acute cholangitis, but does not indicate the presence of dominant stenosis". Please clarify the definition of dominant stenosis in the method section.
Response: We thank the reviewer for this comment. We defined dominant stenosis according to previous studies as stenosis measuring <1.5 mm in the common bile duct or <1.0 mm in the hepatic ducts at cholangiography.
"Dominant stenosis was defined as stenosis measuring <1.5 mm in the common bile duct or <1.0 mm in the hepatic ducts at cholangiography [37,38]." (page 2; l.100)
4. Reviewer: Moderate Language Editing required.
Response: We thank the reviewer for the feedback. Thorough language editing was performed throughout the whole manuscript.
Reviewer 2 Report
This well-structured and written study aimed to investigate the potential role of circulating suPAR as a 68 biomarker in a well-characterized and large cohort of patients with PSC.
Following a well-conducted statistical analysis, this study concludes that suPAR may be a clinically useful diagnostic marker for patients with PSC and could be used for detecting the of stage fibrosis and acute cholangitis episodes in patients with PSC. Moreover, due to the fact that the authors included a cohort of patients with IBD, this study is able to claim that suPAR is not confounded by the presence of disease activity of IBD. As minor suggestions, the authors could detail the information related to the Fibroscan device such as the type of probes used (M / XL, frequency), and the choice of the probe (automatic software/manual). Also, the lack of liver biopsy (the gold standard in fibrosis detection) could be mentioned in the limitations section. Overall, I consider this study to be well written and well designed.
Author Response
We are grateful for the positive feedback.
1.As the reviewer requested we added more details about the probes we used for fibrosis detection via fibroscan in the methods section:
"Detection of fibrosis stage was obtained using liver stiffness measurement by FibroScanMini430 (Echosens; Paris (France)) according to current guidelines [39]. The automatic probe selection tool included in the device's software automatically measured the probe to liver capsule distance and indicated the probe (M or XL) to be used according to the patient's morphology.Cutoff values for fibrosis stages F0-F1, F2, F3, and F4 were <8,7 kPa, ≥8,8 kPa, ≥9,6 kPa, and ≥14,4Pa, respectively." (p.2; l.102-120)
2. Moreover, we added the lack of fibrosis detection to the limitations section:
"The main limitations of our prospective cross-sectional study were the mono-center trial design and the lack of longitudinal measurements for further analysis of suPAR fluctuations during, e.g., endoscopic treatment. In addition, in our study, fibrosis detection was based only on liver stiffness measurements and not on liver biopsy, which is still considered the gold standard for fibrosis detection." (p.14; l.716-720)
This manuscript is a resubmission of an earlier submission. The following is a list of the peer review reports and author responses from that submission.